# In It for the Long Haul: RE-AIM Evaluation of a Preschool Programme Implementing and Maintaining Adult-Initiated Motor Skill Development and Physical Activity across a Two-Year Period

**DOI:** 10.3390/ijerph19052544

**Published:** 2022-02-22

**Authors:** Jonas Vestergaard Nielsen, Thomas Skovgaard, Trine Top Klein-Wengel, Jens Troelsen

**Affiliations:** Research Unit for Active Living, Department of Sports Science and Clinical Biomechanics, University of Southern Denmark, 5230 Odense, Denmark; tskovgaard@health.sdu.dk (T.S.); twengel@health.sdu.dk (T.T.K.-W.); jtroelsen@health.sdu.dk (J.T.)

**Keywords:** preschool, motor skills, physical activity, RE-AIM, process evaluation

## Abstract

Good motor skills (MS) are considered important for children’s social, psychological and physical development and general physical activity (PA) levels. The Motor skill in Preschool study (MiPS) aimed to optimize children’s MS through weekly PA sessions. The aim of this study is to use the RE-AIM framework to report the two-year implementation process of MiPS since the programme’s initiation. Data were collected through a staff questionnaire based on the RE-AIM framework. Data were collected at three months, one year and two years after initiation. Results show that the pedagogical staff believes that the programme promotes MS in children. Implementation measures only showed medium to low fidelity concerning the core element of performing adult-initiated PA sessions with a duration of at least 45 min 4 days a week. The largest barrier was finding the time to plan these PA sessions. Still, the content of the PA sessions achieved high fidelity scores and the programme was deemed suitable for staff’s everyday practice and in alignment with the stated pedagogical goals. The mandatory competence development course was highly valued as strong implementation support. It is notable that there is a large variation in the implementation among the preschools with some struggling more than others.

## 1. Introduction

Good motor skills (MS) are considered important for children’s general physical activity levels as well as their physical, social and psychological development [1,2,3]. Practicing fundamental movement skills are also a necessity in order to create a foundation for more complex movement activities of daily living, recreation and sports in later childhood [1]. Globally, only a small proportion of children (1–5 years old) meet the World Health Organization’s physical activity (PA) guidelines recommending at least 180 min of PA a day, of which one hour should be of moderate to vigorous intensity [4,5]. It is unknown to what degree Danish 1-to-5-year-old children comply with the guidelines; however, a study report that about eight percent of Danish children are experiencing MS difficulties upon reaching school age [6]. Moreover, Danish school-aged children are not physically active according to the stated guidelines from the Danish health authority and WHO [7,8,9]. This suggests that interventions in early childhood are appropriate, while also recommended in order to help promote lifelong healthy PA behaviour [10,11,12]. Furthermore, MS and PA track during the life course, suggesting that intervening in early childhood also seem recommendable in order to promote lifelong healthy PA behaviour [13,14]. Studies have suggested that the implementation of MS and PA interventions in preschool settings could be optimal, to accomplish participation from children from different socio-economic backgrounds and different daily environments (e.g., city and countryside) [15,16,17]. Thus, the need for effective implementation strategies to increase PA levels in the primary years of childhood has been highlighted [10,11,12].

In Denmark, almost all 3-to-6-year-old children in Denmark receive out-of-home childcare [18]. This includes preschool institutions until children turn six when they begin their schooling. Danish preschools are directly linked to school districts but are not an integrated part of the school organization. The pedagogical practice within danish preschools is characterized by a focus on child-to-child relations, dialogue, embodiment and outdoor time [19]. There is an equal focus on children’s independent play and planned learning through activities, resulting in a day-to-day pedagogical practice characterized by a mix of structured activities and unpredictability [15,19].

### 1.1. The MiPS Study

The Motor skill in Preschool study (MiPS) aimed to optimize 3–6-year-old children’s MS through training of both coordination, balance, gross and fine motor challenges, as well as challenging the vestibular, tactile and kinaesthetic senses [17]. To obtain this, four main programme elements were established: (1) Each week, the pedagogic staff had to arrange at least four days of PA sessions with intervals of at least 45 min with a focus on MS development; (2) the PA sessions had to support the training of coordination, balance, and gross and fine motor challenges; (3) pedagogic staff should arrange the PA sessions as tools that can support existing pedagogic goals; and (4) children should be experiencing daily PA with high intensity.

MiPS was initiated in seven public preschools in the municipality of Svendborg, Denmark in January 2017, adding a general focus on MS and PA in their daily pedagogic practice with their 3–6-year-olds [17]. It was the municipality of Svendborg that initially chose to put a focus on MS and PA. However, the programme was developed in collaboration with representatives from the research team, the participating preschools, the Municipality and independent experts in order to ensure a contextual fit and sense of “ownership” of all stakeholders. In order to secure the quality of the programme and support the effective integration of the programme into the institutions’ daily practice, all pedagogic staff had to participate in a competence development course. The course supplied the staff with the tailored knowledge, skills and capacity to deliver the program. Furthermore, to support the implementation, a network of coordinators was established from the participating preschools. The network group consisted of leaders and personnel across all preschools as well as municipal programme managers. Frequent network meetings gave coordinators the opportunity to explain and discuss the implementation in their respective institutions with their peers and programme managers. This offered the opportunity to solve any challenges along the way and aspired towards a mutual inspiration towards the focus on MS and PA training in a preschool setting. 

### 1.2. Process Evaluation

Preschool programmes have been shown to be able to increase children’s PA levels [20,21,22], and MiPS has the potential to gain far-reaching perspectives because the programme is applied at seven institutions, all with different physical conditions and cultures. The Svendborg municipality is also comparable to the rest of Denmark in terms of age distribution, gender and income [17]. However, disseminating behaviour-related programmes into a real-world context such as preschools is often a challenge and there is a need for evaluation with greater attention to the context and the practical implications of programmes [23,24,25]. Process evaluations enhance the transferability of a programme by allowing practitioners and decision-makers to identify and adopt promising programmes that fit their local context. This contributes to a better understanding of the consistency and the internal validity of the programme by, among other things, documenting whether and why (not) the programme was delivered as intended. In general, three important aspects can be identified when performing implementation focused process evaluations: (a) The willingness to adopt the programme, (b) the actual implementation process and (c) the programme’s ability to be maintained over several years [24,25,26]. Furthermore, the knowledge gained from such process evaluations can potentially accommodate the continuing need for strategies on how to translate and disseminate MS and PA programmes into everyday preschool practice [27,28,29,30]. The Reach, Effectiveness, Adoption, Implementation and Maintenance (RE-AIM) framework [31] provides a stepwise approach to process evaluations and allows for a comprehensive evaluation of health-promoting programmes in complex settings [32]. 

Thus, the aim of this study is to enhance the potential transferability of MiPS through a RE-AIM evaluation across a two-year period. In this way, we seek to study the implementation process of MS and PA preschool interventions that are in it for ‘the long haul’, determined to promote lifelong healthy PA behaviour starting in early childhood. 

## 2. Methods

### 2.1. RE-AIM Process Evaluation Framework

In their introduction of the framework, Glasgow et al. argue that, while reach and efficacy might define the impact of a programme, additional attention should be directed towards the adoption, implementation and maintenance dimensions in order to enhance the possible translation and dissemination of programmes [31]. Initially, the adoption dimension relates to the commitment of staff and implementation sites and their decision to install the programme [32]. Secondly, the implementation dimension relates to the extent to which the programme is implemented as intended (fidelity), and the quality of supportive elements [32]. Finally, maintenance relates to the extent to which the elements of the programme, as implemented, were maintained over time and the programme’s ability to become an integrated part of the schools’ daily practice [32]. The recent development of the framework has put emphasis on describing the possible adaptations made to the programme as well as the use of qualitative methods to better understand the implementation context and reasons as to how and why implementation strategies were chosen [32]. Such qualitative aspects are not included in the present RE-AIM evaluation of MiPS, as they have been reported elsewhere [33].

Ultimately, the use of a RE-AIM evaluation across a two-year period, since the programme’s initiation in 2017, enable insights on how the implementation status and staff engagement of MiPS have evolved over time. Each of the RE-AIM dimensions and the outcome measures used in the current study are defined below and in Table 1. 

#### 2.1.1. Reach

The reach dimension represents the characteristics of the preschools that initially became part of the programme when MiPS was launched in 2017. Part of this information has been reported in the background section of the present study, and the MiPS study protocol [17]. Additionally, it is worth highlighting that of the seven participating preschools, four were located in rural areas, two in suburban areas and two in urban areas. The preschools varied in size and had between 3 and 13 pedagogic staff employed.

#### 2.1.2. Effectiveness

Most commonly, effectiveness is evaluated as the impact of a programme on primary outcome measures at the end-user level [25]. Due to the focus on implementation in the current study, the effectiveness of MiPS is, instead, based on the pedagogic staff’s perceptions of the programme’s effectiveness. Staff were asked to what degree they believed the programme had an effect on components related to children’s MS and PA levels, and whether they had noticed an improvement in children’s MS and PA levels since the programme’s initiation. In this way, we seek to capture the pedagogic staff’s assessment of the programme, which is highly relevant in process evaluations at this level, as it has a direct influence on the degree of implementation and maintenance [34,35,36]. The effect evaluation of potential outcomes at the end-user level has been addressed elsewhere [37].

#### 2.1.3. Adoption

Although the seven preschools agreed to become part of the programme, there was no guarantee that the pedagogic staff would adopt the programme. The adoption dimension reports on staff commitment and belief in the programme being relevant in their preschool setting. 

#### 2.1.4. Implementation

This dimension reports on the pedagogic staff’s self-perceived fidelity regarding the main programme elements―including whether the programme was consistently delivered as intended (fidelity). Programme fidelity was measured through staff assessments of the four main programme elements: How many days PA sessions with a focus on MS development was arranged; which kind of MS development staff focused on in their sessions (gross motor challenges, fine motor challenges, coordination exercises and balance exercises); if PA sessions were used to support specified pedagogic goals; and if the children were experiencing daily high intensity PA. Implementation fidelity was collected the first time at the programme’s initiation in 2017, and then again in 2018 and 2019 in order to establish the degree to which MiPS was still ongoing at the preschools. Additionally, the staff were asked to assess four implementation support elements, that were established and highlighted by programme managers: (i) The competence development course, (ii) supplementary programme material available online, (iii) support from the network group and (iv) support from their local colleagues. 

#### 2.1.5. Maintenance

The maintenance dimension reports staff’s self-perceived assessment of the programme’s ability to be ingrained in everyday practice at the seven preschools. Maintenance fidelity was measured one (in 2018) and two (in 2019) years after the programme’s initiation. 

### 2.2. Data Source for the RE-AIM Evaluation

In connection with the process evaluation of MiPS, a questionnaire, assessing the implementation status of the programme, was developed for the pedagogical staff. The questionnaire was developed based on the five dimensions of the RE-AIM framework, with a particular focus on (i) the effects experienced by pedagogic staff, (ii) their adoption of the programme, (iii) the implementation status and (iv) long-term maintenance. Questionnaire data were collected through the online survey programme SurveyXact. Researchers visited each preschool, and all pedagogic staff were asked to fill out the questionnaire on a provided tablet or another mobile device. The staff that were not present at the preschool at the point of data collection were given the opportunity to complete the questionnaire via a printed version and send it to the researchers using an accompanying envelope with pre-paid postage. The questionnaire was collected in the spring of 2017 three months after programme initiation (T1), in the spring of 2018 one year after the implementation (T2) and in the spring of 2019 two years after the implementation (T3). The total number of pedagogic staff across the participating preschools were 66 at T1 (response rate of 81%), 67 at T2 (response rate of 88%) and 66 at T3 (response rate of 95%).

### 2.3. Data Analysis

Questionnaire data were downloaded in a csv-format from the electronic survey program SurveyXact and imported into Stata version 16 to perform the analysis. Initially, the dataset was cleaned for any obvious typing mistakes. Due to the paucity of responses, the values “partially agree” and “agree” were collapsed, and the value “partially disagree” was collapsed with the value “disagree” in the analysis. Thus, two categories were used in the analysis: (i) Agree and (ii) disagree. Finally, descriptive statistics and proportions on implementation fidelity were produced. Pearson chi-square analysis was employed to test for differences at a 0.05 significant level between T1, T2 and T3 as well as variations between the preschools at each timepoint.

### 2.4. Ethics

The questionnaire was anonymous and did not collect personally identifiable information such as name, address or civil registration data. In addition, all PE teachers in the study received an information letter to inform them about the study. A passively informed consent procedure was used, automatically including the pedagogic staff unless they actively withdrew consent. It was made clear and easy for the pedagogic staff to reject participation or withdraw from the study without providing any explanation. To withdraw from the study, participants had to inform the research team through the given contact information in the information letter. This procedure is in accordance with Danish regulations in anonymous and low-risk research. The study was approved by the Regional Committees on Health Research Ethics for Southern Denmark (S-2015-0178) as well as by the Danish Data protection Agency (2015-57-0008).

## 3. Results

### 3.1. Effectiveness

At T1, 89% of the pedagogic staff indicated that the programme promotes preschool children’s MS (Table 1). At T2, this number decreased to 76%, and at T3, 86% of the pedagogic staff indicated that the programme promotes the MS of children. When looking at the individual preschool institutions, the results show some variation regarding the experience of the programme being effective, ranging from 78 to 100% at T1 and 57–100% at T3. Overall, the results show that more than nine out of ten of the pedagogic staff believed the programme promotes MS (Table 2).

### 3.2. Adoption

More than eight out of ten of the pedagogic staff believed that the programme is relevant and were motivated to be part of the programme across T1-T3 (Table 3). Results show that staff motivation decreased from 98 to 89% during T1-T3 and variations across preschool institutions show that the lowest response to motivation drops from 89% at T1 to 57% at T3. When asked if they have sufficient time to plan activities that support MS learning, 69% of the pedagogic staff agreed at T1, 66% at T2 and 65% at T3. Regarding having sufficient time, the individual variation across preschools dropped from 22–100% at T1 to 0–92% at T3. As for the PA session’s ability to support pedagogic goals, 93% agreed at T1 and 95% agreed at both T2 and T3 (Table 1). When looking at preschool variations, there was generally high agreement, yet a small decrease at some institutions was observed, from 89–100% at T1 to 71–100% T3.

### 3.3. Implementation

The programme had four requirements for full implementation: (1) At least four days of PA sessions with intervals of at least 45 min with a focus on MS development; (2) the PA sessions had to support the training of gross motor challenges, fine motor challenges, coordination exercises and balance exercises; (3) PA sessions should support existing pedagogic goals and (4) children should achieve daily PA with high intensity. The fidelity of these requirements was self-assessed by the pedagogic staff. The results are presented in Table 4 and Table 5. At T1, 65% of the children received PA sessions with a focus on MS more than 4 days a week. Furthermore, 48% of the sessions lasted more than 45 min (Table 4). At T3, 70% received more than four sessions per week, and 54% of the PA sessions lasted more than 45 min. Regarding PA sessions of at least a 45 min duration, results showed large individual preschool variations, ranging from as low as 0% at some preschools to 100% at others (Table 4). 

Table 5 presents the fidelity of programme requirements regarding the content of PA sessions. Gross MS exercises showed the best fidelity measures in the range of 96–94% across T1–T3, whereas balance exercises showed the lowest fidelity ranging 67–56% (Table 5). At T1, coordination exercises showed the second-best fidelity (91%) followed by fine MS exercises. However, at T3, fine MS exercises showed the second-best fidelity (89%) as coordination exercises dropped to 79%. For gross MS, fine MS and coordination exercises, individual preschool variation is between 50 and 100%, whereas variations for balance exercises in some cases range from 17 to 100% (Table 5).

When asked whether the children achieve daily PA with high intensity, and if the PA sessions supported general pedagogic goals, more than nine out of ten of the pedagogic staff agreed across T1–T3 (Table 4).

### 3.4. Implementation Support

The pedagogic staff took part in a competence development course and had access to online supplementary material to support further programme implementation. A network group was established in which experiences and current implementation processes could be shared. The network group consisted of managers and personnel across all preschools as well as municipal programme coordinators. Lastly, pedagogic staff were highly encouraged to plan and discuss the implementation process with colleagues from their own institution. Results show that eight out of nine pedagogical staff found the competence development course and the discussions with local colleagues valuable and useful for their ability to implement the programme (Table 6). Around one-third found that the online supplementary material strengthened their ability to implement the programme, yet there are large variations between the different preschools ranging 0–100% at T1. At T1, 51% of the pedagogic staff found that discussion within the programme network is a usable component for their implementation efforts, while this figure is reduced to 10% at T3.

### 3.5. Maintenance

As seen in Table 4 and Table 5, the implementation rate of programme requirements only varied to a small degree over time. Three out of the four fidelity measure points, concerning the content of the planned PA sessions, decreased from T1 to T3, with the largest being PA activities containing coordination exercises (91 to 79%). Regarding fidelity of the structural programme requirements, three out of four increased and one was static from T1 to T3. 

In the survey collected at T2 and T3, the pedagogic staff were asked about three maintenance-related indicators. As seen in Table 7, 78% of the pedagogic staff found that MS and PA development has become an integrated part of their daily practice at T2 and 86% at T3. When asked if the programme has enhanced their motivation regarding MS and PA development, 83% agree at T2 and 92% agree at T3. Finally, 83% agree that they have been able to adapt the program to their daily practice at T2 and 87% agree at T3. 

## 4. Discussion

The aim of this study is to evaluate MiPS and enhance the programme’s potential transferability using the RE-AIM framework. Based on questionnaire data, following the programme over a two-year period, we assessed to what degree staff engagement as well as programme implementation and maintenance was attained ‘in the long haul’. These results will be discussed in the following sections.

### 4.1. Effectiveness 

Previous health promotion initiatives in a preschool setting have reported increased PA, but overall, the results are rather inconsistent [38,39,40]. Programmes implemented in a real-world setting, as it is the case for MiPS, face a number of challenges potentially affecting their outcomes [39]. In the current study, the pedagogic staff’s belief in and self-assessment of the change in MS was used as an indicator of effectiveness. Almost nine out of ten pedagogues stated that the programme promoted MS in the children at T1, and, most noticeably, most of them still noted this effect two years into the programme at T3. All staff were thoroughly introduced to the programme through the available online materials and their participation in the competence development course. Still, we do not know whether the staff fully recognize all aspects of MS and are able to assess any and all potential outcomes of the programme. It is recognized that the staff’s self-assessment might not be an accurate measure for any actual change in the children’s MS development. Yet, in the aim of the current study, the staff’s solid belief in the effectiveness of MiPS is evident and constitutes an important marker for their dedication and motivation in implementing the programme [30,41,42,43].

### 4.2. Adoption

The pedagogic staff’s willingness to participate in a given programme is an important aspect of a process evaluation [44]. The adoption rates concerning the relevance and motivation of being part of MiPS drops 5–9% from T1 to T2 but are generally high. This is a positive finding as the attitude and values of the pedagogic staff are also considered to relate to children’s overall PA level within the preschool settings [15,16,45]. 

The implementation and further roll-out of programmes like MiPS, with an added focus on MS and PA development, are often challenged by resistance (fair or not) from, for instance, members of staff that do not agree on the strengthened focus on e.g., physical activity [42]. However, the results from this study show that more than nine out of ten of the pedagogic staff agree that the PA sessions assisted their pedagogic work by helping them achieve pedagogic goals. In general, the literature shows that programmes like MiPS are more implementable when they can be adjusted for professional competencies and preferences among key implementation agents, such as pedagogical staff [41,46,47]. Still, results also show that the staff have difficulty finding the time to plan the required MS development sessions. The average adoption rates regarding sufficient time are mid-range to good. Furthermore, when looking at the maximum variation between preschools, it is clear that some are struggling. This might be due to low prioritization by local preschool leaders or a general lack of organizational support (e.g., from the municipal programme mangers) to follow through on implementation decisions [26,30,33,43].

### 4.3. Implementation

Two of the main components of the MiPS programme were the 45 min PA sessions with a focus on MS at least four days a week. Implementation measures show only medium to the high fidelity of these sessions being performed at least four days a week, and medium to low fidelity concerning the sessions having a duration of at least 45 min. Especially, this requirement showed large variations between preschools. Such variations between individual implementations sites have also been reported in other Danish PA preschools studies [15] and are often suggested to relate to a difference in the social and organizational environment [16]. The large variation in fidelity could challenge the effectiveness of MiPS as the implementation of core elements is generally considered important in order for programmes to produce expected impacts [26,46]. Despite the lack of strong fidelity measures in the planning and duration of PA sessions, the content of the arranged PA sessions achieved fairly high fidelity scores. This could indicate that the pedagogic staff has adopted a focus on MS development, but as noted earlier, is having difficulties finding the time to follow through on the requirement of 45 min. 

An important aspect of the PA sessions was to support the established pedagogic work of the staff, making the programme an add-in (oppose to an add-on) assignment in their daily work. When asked if the staff used PA sessions to support pedagogic goals, almost all agreed. Such alignment with the existing practice is key in order to increase a sense of relevance, ownership and general belief in a programme [36,46,48]. In addition, there is a general consensus that investment in skill development of the front personnel facilitates the capacity for change and securing implementation fidelity [26,49,50]. Results from the present study show that the professional development courses also were highly valued by the pedagogic staff across T1 to T3. Nevertheless, it is worth noting that no matter how well a professional development course is constructed, it is often not, by itself, enough to change practitioners’ behaviour [49,51]. Establishing a foundation for successful implementation needs additional structures to support front personnel practice and ensure programme delivery [48,49,51]. The results show that among the supporting structures established in MiPS, the staff most valued having a professional development course and a general collaborative implementation attitude in their local preschool through collegial discussions. In order to support the implementation process, MiPS also established a network group and a set of online programme material, which both showed a low value in staff’s implementation of the programme. This was rather surprising as a local collaborative network has been especially recommended in order to both increase staff relatedness as well as their engagement in implementing a PA programme [36,52,53]. 

### 4.4. Maintenance

Implementation fidelity across T1 to T3 is fairly stable, and in some cases, actually increased. In addition, maintenance results show that more than eight out of ten pedagogues find the programmes focus on MS and PA development as motivating. The results seem to suggest that core elements of MiPS have been maintained as part of daily practice. When directly asked, the majority of staff also agrees that they have been able to adapt the programme and that the programme is now embedded in their daily practice. Adaptation to the local setting empowers local decision making and the sense of ownership, which supports the instalment of the programme [26,46,54]. In MiPS, staff were able to apply a local adaptation of the programme, which could have been central as each of the preschools had different sizes, facilities and local resources [17]. 

### 4.5. Strengths and Limitations

One of the strengths of the current study is the continuous collection of implementation information over two years and documenting the process from the programme’s initiation in 2017. Through the RE-AIM framework, the study has an extensive focus on the implementation processes. Together with a focus on the experiences of the local staff, this most likely has enhanced the relevance and possible transferability of the programme for other preschools. This is relevant due to the notion that the most efficient way to establish an effective health promotion practice is to learn from programmes we know ‘work’ in practice [55,56]. The results regarding strategies and influence on the implementation were discussed and mainly supported by the international literature. Still, the transparency and applicability of the knowledge produced in the study should be viewed in light of it being based on a limited sample in a Danish preschool context. In a Danish context, the Municipality of Svendborg is close to an average Danish municipality on a number of relevant aspects [17], thus enhancing the relevance and possible dissemination of the programme to other Danish preschools [54,57]. Furthermore, the programme was initiated by the municipality, thus not directly for the purpose of research, which evidently increases the likelihood that similar initiatives could be implemented elsewhere in Denmark and possibly in other countries.

It is also important to address that the representation of pedagogues through only questionnaires provides a need for further qualitative knowledge regarding the importance of local leadership as well as staff and parents’ perspective when implementing these real-world programmes [32]. Such elaborative qualitative data could also explore the noted differences between the investigated preschools, as well as the importance of local leadership when implementing and maintaining MS and PA interventions in this particular setting. Finally, it should be noted that all seven preschools in the study chose to become a part of the project. Thus, the focus on MS and PA might have been in alignment with their individual preschool’s existing values and priorities. This could imply that lesser motivated preschools, without any prior experience in MS and PA development, might experience other, more extensive implementation challenges.

### 4.6. Implications for Practice 

Firstly, professional development courses in combination with what can be called a collaborative implementation attitude are highly valued by the pedagogic staff and support their ability to implement the programme containing more PA with a focus on MS in a Danish preschool setting. The results also indicate that investment in the training of pedagogues has supported the implementation positive across a two-year period. This type of training could improve pedagogues’ motivation, skill level and help secure a feeling of competence to change their daily work—with a focus on MS and PA. 

Furthermore, it could be beneficial for elements in the programme to be flexible for local adaptation while also assisting the pedagogical work and aiding staff to achieve pedagogical goals. Even though there needs to be some sense of frame in which the pedagogic staff can identify core elements that have to be implemented, it should be possible to make the programme an add-in assignment where it can be aligned with existing practice and thereby increase staff motivation and belief in the programme. 

Lastly, the current study highlights that it is important to prioritize sufficient time when implementing a new MS and PA programme in preschools. In the case of MiPS, even though the pedagogic staff adopted a focus on MS development, some had difficulty finding the time to plan the required MS development sessions and follow through on the requirement of 45 min a day. The organizational structure and local managers should be considered important stakeholders in balancing such time issues so that the pedagogic staff are able to meet the programme requirements. 

## 5. Conclusions

The current study explored the implementation of MiPS, a Danish MS and PA preschool programme, through a RE-AIM evaluation. The aim was to enhance potential transferability, to ultimately enable practitioners and decision-makers to identify and adopt promising elements that fit their local context, aiding the future promotion of lifelong healthy PA behaviour in early childhood. 

The results show that the pedagogic staff believed that MiPS promoted MS in preschool children. Implementation measures only showed medium to low fidelity concerning the core element of performing adult-initiated PA sessions with a duration of at least 45 min 4 days a week. Despite the lack of strong fidelity measures in the planning and duration of PA sessions, the content of the arranged PA sessions achieved fairly high fidelity scores. Moreover, core programme elements were seen as suitable for everyday practice, and the pedagogical staff used the PA sessions to support their pedagogical work and achieve pedagogical goals. The staff participated in a professional development course, which, together with support from local colleagues, was highly valued and contributed to the staff´s ability to implement the programme. Variation between the preschools was found, yet the pedagogical staff generally deemed the programme valuable in promoting MS among children and found it relevant and motivating to be part of the programme. As a barrier for implementation, some off the pedagogical staff had difficulties finding the time to plan the required MS sessions as well as running sessions at least four days a week and sessions with a duration of at least 45 min. This study provides valuable insight into the potential transferability of the MiPS programme promoting PA sessions with a focus on optimizing children’s MS in Danish preschools across a two-year period. Still, further research is needed, both international and in a broader Danish perspective, focusing on outcome measures at the end-user level accompanied by comprehensive qualitative insights on the implementation of MS and PA in preschools that are committed long term.

## Figures and Tables

**Table 1 ijerph-19-02544-t001:** Outline of the RE-AIM dimensions and outcome measures used in the study.

Dimension	Definition	Outcome Measures
Reach	Refers to the proportion and representativeness of eligible preschools willing to participate in the study.	- This dimension is mainly reported elsewhere [17], and only minor supplements are mentioned in this paper
Effectiveness	The staff’s perception of the degree to which the PA programme components influenced the children’s MS.	- Pedagogic staffs’ perceived effect of the programme on the children’s MS
Adoption	The commitment of pedagogic staff on the participating preschools regarding their decision to install the programme.	- Staff’s belief in the programme being relevant - Staffs’ ability to use PA to achieve pedagogic goals
Implementation	The extent to which preschool staff implemented the programme as intended and their assessment of the planned implementation support elements.	- Degree of programme elements that were delivered as designed (fidelity) - Staff’s assessment of planned implementation support elements
Maintenance	The programme’s ability to become an integrated part of daily practice.	- Programmes ability to motivate staff and become part of their daily practice.

**Table 2 ijerph-19-02544-t002:** Effectiveness regarding the promotion of MS based on the experiences of pedagogic staff.

	I Experience That the Programme Promotes the Children’s MS		I Believe the Programme Promotes the Children’s MS	
	**Agree %**	**School Range %**	**Agree %**	**School Range %**
T1 (*n* = 66)	89%	78–100%	98%	77–100%
T2 (*n* = 67)	76%	33–100% *	98%	50–100%
T3 (*n* = 66)	86%	57–100%	92%	71–100%

* Significant difference between preschools in chi-Square tests: *p* value ≤ 0.05.

**Table 3 ijerph-19-02544-t003:** Adoption of the programme in pedagogic staff.

	I Believe That the Programme Is Relevant		I Am Motivated in Being Part of the Programme		I Have the Sufficient Time to Plan Sessions That Support MS Development		The PA Sessions Have Helped Me Achieve Pedagogic Goals	
	**Agree %**	**School Range %**	**Agree %**	**School Range %**	**Agree %**	**School Range %**	**Agree %**	**School Range %**
T1 (*n* = 66)	94%	58–100%	98%	89–100%	69%	22–100% *	93%	89–100%
T2 (*n* = 67)	86%	67–100%	80% **	33–100% *	66%	0–100% *	95%	83–100% *
T3 (*n* = 66)	89%	57–100%	89% **	57–100%	65%	0–92% *	95%	71–100%

* Significant difference between preschools in chi-Square tests: *p* value ≤ 0.05, ** Significant change from T1 in chi-Square tests: *p* value ≤ 0.05.

**Table 4 ijerph-19-02544-t004:** Implementation fidelity of core programme requirements, based on the assessment of pedagogic staff.

	The Children Are Achieving PA Sessions with a Focus on MS at Least 4 Days a Week		The Children Are Achieving PA Sessions of 45 min with a Focus on MS at Least 4 Days a Week		The Children Achieve Daily PA with High Intensity		I Have Used PA Sessions to Support My Pedagogic Work	
	**Agree %**	**School Range %**	**Agree %**	**School Range %**	**Agree %**	**School Range %**	**Agree %**	**School Range %**
T1 (*n* = 66)	65%	33–100%	48%	22–100% *	98%	89–100%	96%	89–100%
T2 (*n* = 67)	68%	33–100% *	46%	0–100% *	93%	67–100%	98%	75–100% *
T3 (*n* = 66)	70%	43–100%	54%	25–100% *	98%	90–100%	97%	86–100%

* Significant difference between preschools in chi-Square tests: *p* value ≤ 0.05.

**Table 5 ijerph-19-02544-t005:** Implementation fidelity of the content of PA sessions, based on the assessment of pedagogic staff.

	During the Last Two Weeks I Have Arranged PA Activities with Gross MS Challenges		During the Last Two Weeks I Have Arranged PA Activities with Fine MS Challenges		During the Last two Weeks I Have Arranged PA Activities with Coordination Challenges		During the Last Two Weeks I Have Arranged PA Activities with Balance Challenges	
	**Agree %**	**School Range %**	**Agree %**	**School Range %**	**Agree %**	**School Range %**	**Agree %**	**School Range %**
T1 (*n* = 66)	96%	75–100%	81%	57–100%	91%	67–100%	67%	25–100% *
T2 (*n* = 67)	97%	75–100%	78%	50–100%	85%	50–100%	56%	17–100%
T3 (*n* = 66)	94%	83–100%	89%	75–100%	79%	50–100%	56%	17–100% *

* Significant difference between preschools in chi-Square tests: *p* value ≤ 0.05.

**Table 6 ijerph-19-02544-t006:** Implementation support assessed by pedagogic staff.

	The Competence Development Course Has Strengthened My Ability to Implement the Programme		The Supplementary Material on the Programme Website Has Strengthened My Ability to Implement the Programme		Discussions with the Programme Network Group Has Strengthened My Ability to Implement the Programme		Discussion with Local Colleagues Has Strengthened My Ability to Implement the Programme	
	**Agree %**	**School Range %**	**Agree %**	**School Range %**	**Agree %**	**School Range %**	**Agree %**	**School Range %**
T1 (*n* = 66)	87%	67–100%	39%	0–100% *	41%	15–67%	96%	89–100%
T2 (*n* = 67)	78%	50–100%	34%	0–70% *	22% **	0–43%	88%	50–100% *
T3 (*n* = 66)	83%	58–100%	32%	0–50%	10% **	0–50% *	89%	0–100%

* Significant difference between preschools in chi-Square tests: *p* value ≤ 0.05, ** Significant change from T1 in chi-Square tests: *p* value ≤ 0.05.

**Table 7 ijerph-19-02544-t007:** Maintenance-related indicators assessed by the pedagogic staff.

	MS and PA Development Has Become a Natural Integrated Part of My Daily Practice		Being Part of the Programme Has Enhanced My Motivation Regarding MS and PA Development		It Has Been Possible to Adapt the Programme to My Daily Practise	
	**Agree %**	**School Range %**	**Agree %**	**School Range %**	**Agree %**	**School Range %**
T2 (*n* = 67)	78%	50–100%	83%	67–100%	83%	50–100%
T3 (*n* = 66)	86%	57–100%	92%	57–100% *	87%	43–100% *

* Significant difference between preschools in chi-Square tests: *p* value ≤ 0.05.

## Data Availability

The data presented in this study can become available on request from the corresponding author. The data are not publicly available due to legal and privacy issues.

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
