# Peer review of "In It for the Long Haul: RE-AIM Evaluation of a Preschool Programme Implementing and Maintaining Adult-Initiated Motor Skill Development and Physical Activity across a Two-Year Period"

_ijerph, 2022, doi:10.3390/ijerph19052544_

Round 1
Reviewer 1 Report
Although the research topic is fascinating, I recommend to improvement article with some comments as below:
1.More description for experiments should be done. Table 1, it is confusing for readers to understand how those themes and sub-themes came up with, based on what framework/model, and why these definitions and outcomes were chosen and emphasised.
2.The method can be explained in a clearer way. Try to explain the theory more detailed in discussion. Please add necessary references.
3.I learn few new things about The RE-AIM Framework was developed in the late 1990s. In particular, we learn nothing specific about how these approaches could be, or are currently.
4.Its results are not well presented. I think researchers need to be precise about the scope of their findings and be more cautious about the generalizations. Because your study is based on a limited sample, and all samples are from a single country, Denmark.
5. The conclusion section should not only repeat the results already presented in the discussion section, but should more clearly indicate the study's innovation and limitations
Author Response
Reviewer 1
Although the research topic is fascinating, I recommend to improvement article with some comments as below.
Authors’ comment: Thank you, we highly appreciate your comments. We have strived to address each of your concerns below. We do find that it has helped improved the manuscript. All the revisions are marked in yellow in the revised manuscript.
Introduction:
- More description for experiments should be done. Table 1, it is confusing for readers to understand how those themes and sub-themes came up with, based on what framework/model, and why these definitions and outcomes were chosen and emphasised.
Authors’ comment: Please find that we have added information regarding the RE-AIM framework (line 113-128) in order to make the chain of reasoning more transparent regarding the information in table 1.
- The method can be explained in a clearer way. Try to explain the theory more detailed in discussion. Please add necessary references
Authors’ comment: We are not completely sure that we understand your comment correctly. We have strived to add information (line 113-128) and discussions (line 424-435) on the method and the RE-AIM in the manuscript.
- I learn few new things about The RE-AIM Framework was developed in the late 1990s. In particular, we learn nothing specific about how these approaches could be, or are currently.
Authors’ comment: Again, we refer to the added text in line 113-128 and hope that this is sufficient information. Specifically, we have added updated information and citations regarding the RE-AIM framework (Galsgow et al 2019; RE-AIM Planning and Evaluation Framework: Adapting to New Science and Practice With a 20-Year Review)
- Its results are not well presented. I think researchers need to be precise about the scope of their findings and be more cautious about the generalizations. Because your study is based on a limited sample, and all samples are from a single country, Denmark.
Authors’ comment: Thank you for this comment. We have made some revisions throughout the results, also including section 4.5 ‘implication for practice’, in order to both highlight the limited sample and the Danish context as well as adjust the language to be less generalizing.
- The conclusion section should not only repeat the results already presented in the discussion section, but should more clearly indicate the study's innovation and limitations
Authors’ comment: Please find that we have made revisions in section 5 ‘conclusion. We hope that these revisions are satisfying.

Reviewer 2 Report
In the introduction of the study it should be provided more theoretical backgrounds regarding the subject in order to support the research.
Author Response
Reviewer 2
- In the introduction of the study it should be provided more theoretical backgrounds regarding the subject in order to support the research.
Authors’ comment: Thank you for this comment. Please find that we have made revisions and added information in section 1 ‘Introduction’.

Reviewer 3 Report
This paper aims to assess the feasibility of increasing physical activity in preschool children through established and validated programme.
In the introduction, I don't understand whether the children are in a day care centre or at home before entering school. It would be good to mention the age of the children when they enter school and the starting age of this programme because the educational system is not the same in the different European countries.
So I don't understand who is "the pedagogical staff", physical activity teachers or generalist teachers : could you precise ?
I also don't see the duration of such a programme for a child (one year, two years, three years ?) and therefore don't understand the temporality of the questionnaires repeated each year with the professionals
Author Response
Reviewer 3
This paper aims to assess the feasibility of increasing physical activity in preschool children through established and validated programme.
Authors’ comment: Thank you for your comments, they are relevant and have surely strengthened the quality of the manuscript. All the revisions are marked in yellow in the revised manuscript
- In the introduction, I don't understand whether the children are in a day care centre or at home before entering school. It would be good to mention the age of the children when they enter school and the starting age of this programme because the educational system is not the same in the different European countries.
Authors’ comment: This is a very relevant comment. Please find that we have added information on the danish context including age groups and how preschools are organized (line 49-52).
- So I don't understand who is "the pedagogical staff", physical activity teachers or generalist teachers : could you precise ?
Authors’ comment: In relation to the previous comment, we have also added information on pedagogic staff and their daily practice in line 52-56. We hope that this information in satisfying.
- I also don't see the duration of such a programme for a child (one year, two years, three years ?) and therefore don't understand the temporality of the questionnaires repeated each year with the professionals
Authors’ comment: Again, we hope that the added information on the danish preschool context is sufficient, and we hope that the added paragraph (lines 49-56) heightens the transparency regarding the children’s course in the programme. Regarding the temporality of the questionnaire, this mainly relates to the manuscripts core focus on implementation and maintenance of MiPS and the mechanisms at play in order for such programmes to be able to promote lifelong healthy PA behaviour starting in early childhood.
A more direct focus on the children (and outcome) can be found in the articles below:
- Hestbaek, L.; Andersen, S. T.; Skovgaard, T.; Olesen, L. G.; Elmose, M.; Bleses, D.; Andersen, S. C.; Lauridsen, H. H., Influence of motor skills training on children's development evaluated in the Motor skills in PreSchool (MiPS) study-DK: study protocol for a randomized controlled trial, nested in a cohort study. Trials 2017, 18, (1), 400.
- Hestbaek, L.; Vach, W.; Andersen, S. T.; Lauridsen, H. H., The Effect of a Structured Intervention to Improve Motor Skills in Preschool Children: Results of a Randomized Controlled Trial Nested in a Cohort Study of Danish Preschool

Round 2
Reviewer 3 Report
They responded to all the comments and corrected the document accordingly